# Differential Gene Expression in Red Imported Fire Ant (*Solenopsis invicta*) (Hymenoptera: Formicidae) Larval and Pupal Stages

**DOI:** 10.3390/insects9040185

**Published:** 2018-12-05

**Authors:** Margaret L. Allen, Joshua H. Rhoades, Michael E. Sparks, Michael J. Grodowitz

**Affiliations:** 1USDA-ARS Biological Control of Pests Research Unit, National Biological Control Laboratory, Stoneville, MS 38776, USA; michael.grodowitz@ars.usda.gov; 2USDA-ARS Invasive Insect Biocontrol and Behavior Laboratory, Beltsville, MD 20705, USA; jrhoades@udel.edu (J.H.R.); michael.sparks@ars.usda.gov (M.E.S.)

**Keywords:** fire ant brood, digestion, transcriptome, RNA-Seq, differential expression

## Abstract

*Solenopsis invicta* Buren is an invasive ant species that has been introduced to multiple continents. One such area, the southern United States, has a history of multiple control projects using chemical pesticides over varying ranges, often resulting in non-target effects across trophic levels. With the advent of next generation sequencing and RNAi technology, novel investigations and new control methods are possible. A robust genome-guided transcriptome assembly was used to investigate gene expression differences between *S. invicta* larvae and pupae. These life stages differ in many physiological processes; of special importance is the vital role of *S. invicta* larvae as the colonies’ “communal gut”. Differentially expressed transcripts were identified related to many important physiological processes, including digestion, development, cell regulation and hormone signaling. This dataset provides essential developmental knowledge that reveals the dramatic changes in gene expression associated with social insect life stage roles, and can be leveraged using RNAi to develop effective control methods.

## 1. Introduction

*Solenopsis invicta* Buren (Hymenoptera: Formicidae), commonly referred to as the red imported fire ant (RIFA), is native to central South America and has been introduced to North and Central America, the Caribbean, Asia, Australia and New Zealand (http://www.cabi.org/isc/datasheet/50569, accessed 1 August 2017). One of the earliest introductions was to the Southern United States. First reported in Mobile, Alabama in 1930, invasive fire ants, including *S. invicta*, have since spread across most of the southern United States and may expand as far north as Virgina and west to Oregon [1,2]. Myriad ecological, agricultural and economic impacts have been reported [1,3,4]. In its native range, *S. invicta* is kept in check by predators, pathogens, and competition with other ant species, which are not present in the invaded areas. Development of safe and practical control mechanisms for *S. invicta* is important, both to protect native species and habitats, as well as to prevent economic impacts to agriculture and businesses.

Several large-scale control attempts using the broad-scale insecticides Heptachlor, Dieldrin and Mirex were unsuccessful, resulting in extensive off-target effects impacting many species (reviewed in [1,5,6]). Ecologically friendly control methods are needed for *S. invicta*, which requires an intimate knowledge of important physiological processes. The *S. invicta* genome has been sequenced and is available in public databases [7]. Genomic and genetic data from ants have been used to study development [8,9], social evolution [10,11,12], social behavior [13,14,15,16,17], reproduction [18,19] and chemical communications [20,21]. A microarray and cDNA library were deposited [8] for the express purpose of differential expression studies. While these data resources have been extraordinarily useful for annotation [22] and broad comparisons relating to social form [19,23,24] and evolution [25], some studies have required and utilized further transcriptome sequencing. For example, differential gene expression studies revealed 22 genes that differ between alate virgin and dealate mated queen brains in *S. invicta* [26]. Because our study aim was to collect precise expressed sequences from the same colony specimens that we intended to use for gene disruption, we prepared and compared transcriptomes from our target populations.

*Solenopsis invicta* is an eusocial insect; the ants live in large colonies within subterranean nests with interdependent division of labor among individuals of all life stages and multiple castes. As with all holometabolous insects, life begins as an egg (Figure 1A), which hatches and then feeds and grows rapidly during the larval (Figure 1B) stage. Once large enough, the larvae void gut contents to enter pupation as a pre-pupa, then complete a molt to become pupae. The pupa is initially white (Figure 1C), and darkens (Figure 1D) as it approaches the adult molt and then finally emerges as an adult (Figure 1E). Of particular importance is the role of larvae in providing the colony with a communal gut; larvae have been described as “a protein digestive organ for the colony” [27]. Adult ants lack digestive proteases and therefore do not digest protein efficiently [28,29]. Fire ant colonies primarily process and digest proteins via late-stage larvae [30,31]. Unique mouthparts [32,33], a body shape and setae that form a food basket [1], salivary secretions [28,34], and filtration structures [35] make *S. invicta* larvae uniquely capable of digesting complex molecules. Protein-rich food items are brought to the larvae for digestion and subsequent regurgitation, a process called stomodeal trophallaxis [36]. While most adult workers do not have large protein demands, the queen requires an abundant, uninterrupted supply of protein for egg laying. The regurgitated food is collected by the nurse ants, who retain some but feed the majority to the queen [37]. This “child labor” [36] leverages normal larval digestive processes, allowing other members of the colony to forego energy and resource expenditures that are required to digest protein-rich food items. The changes in gene expression associated with the juvenile role of fire ants have not been examined in detail.

Gene disruption methods including RNA interference (RNAi) have been widely used to study genes. Systematic, high throughput RNAi tools for identification of pest control target genes are being developed in a model insect, the beetle *Tribolium castaneum* [38]. The basis of RNAi is delivery of double stranded RNA to an organism resulting in decrease of very specific gene transcripts (reviewed in [39,40,41]). Specifically in ants, RNAi has been used to decrease *S. invicta* queen egg production [42] and to impair worker survival [43] (and see [44]), and to disrupt chemical communication [44,45]. Genetic pest control for insects using RNAi is a complicated proposal [39,40,41,46], but has led to one viable commercial pest control strategy to combat corn rootworm, a beetle pest [47,48]. In the context of a social insect such as imported fire ants the complexity of using RNAi as a control strategy is increased. Not only does the double-stranded RNA need to pass from the environment to the cells of the insect, but it needs to pass to the entire ant colony. A product designed as effective to adult workers, either by causing death or confusion, might remove only the expendable foragers contacting the product and thus not spread to the colony; that is, the product would not enter the colony if disruption of the gene target was lethal to adult workers. A product detrimental to the colony digestive or developmental systems is arguably a more logical choice. The first step in developing such a genetic ant control product is to find genes critical to the larval digestive system or larval vitality, and secondarily the larva to pupa developmental system. Because RNAi is based on precise gene sequences, the ideal source for those sequences should be the nearest relatives to the study organisms. Here, we survey transcriptomes of three established laboratory colonies with an aim to identify differentially expressed genes between the fourth (final) larval and early (white) pupal stages. Because these stages are both juvenile and immobile, they could be separated alive from the colonies and sorted after adult removal. The larva stage represents the colony gut, and the pupa is a non-feeding stage, and thus represented an oppositional comparison. We placed emphasis on transcripts related to important physiological processes that differed between these two juvenile life stages, with an aim to identify candidate gene targets for RNAi applications. This effort was not strictly hypothesis-based but intended to produce a data source for further studies. We anticipated finding numerous potential RNAi targets related to the digestive role of the larval stage. We expected to find transcripts associated with the larval “communal gut” function: complex molecule digestion enzymes such as proteinases, lipases, and carbohydases in the larva transcriptome. We also expected to find larva transcripts associated with storage, detoxification, and membrane transport. The specific categories examined were digestion, nutrient storage, xenobiotic detoxification, muscle, cuticle, neural development, cell regulation and signaling, hormone signaling, fatty acid metabolism, immunity, and caste determination. Caste determination transcripts were expected to exhibit little variance because our samples were juvenile and worker caste.

## 2. Materials and Methods

### 2.1. Colony Origin and Rearing

*Solenopsis invicta* samples were collected in Washington County, MS (33°27′48.3″ N 91°01′29.0″ W) following a flash flood in March 2016. Numerous rafting colonies were visible in a flooded pasture, and four individual, isolated, large clusters of floating ants—presumed to represent colonies—were collected and transferred to laboratory facilities at the National Biological Control Laboratory in Stoneville, MS. To replicate temperate underground conditions, insect colonies were reared in a controlled environment room at a constant 26.5 °C and 50%RH, with photoperiod limited to only required rearing maintenance time, which was normally less than 10 h per week. Colonies were provided with glass tubes for brood chambers and diet consisting primarily of frozen insects and water ad libitum. Methods for housing and handling ant colonies were otherwise similar to those described [49,50].

### 2.2. Sampling and RNA Extraction

Colonies were in growth stage when sampled; reproductive individuals were not present as larvae, pupae nor winged adults (alates). Each colony contained more than one queen and was therefore presumed to be polygyne, typical of the geographical region from which the colony samples were collected. In total, six samples were taken from three colonies. Insects were immobilized by exposure to carbon dioxide, and adults were removed and replaced in the colony. Late stage worker larvae (L4) were selected based on size and the presence of darkened mandibles and visible food in the midgut. For each of the three colonies, 20 individuals in the fourth instar larval stage (Figure 1B) and 20 individuals from the early (white, but with visible adult morphology, Figure 1C) pupal stage (WP) were collected. For each sample of 20 juvenile insects, individuals were pooled for mRNA extraction, and once isolated immediately added to extraction buffer and homogenized. Thus, we had paired samples, one larva sample (20 larvae) and one pupa sample (20 pupae), from each of three colonies, which for analysis were treated as three biological replicates of two juvenile stages.

Total RNA extraction from each of these six samples followed the USB PrepEase^®^ (USB Corporation, Cleveland, OH, USA) protocol per manufacturer instructions for animal tissues. The on-membrane DNA digestion step was included. Concentration and relative purity were measured using a NanoDrop^®^ ND-1000 (Thermo Fisher Scientific, Wilmington, DE, USA). Total RNA used for sequencing was of uniform high quality (OD ratios of 2.0 or better for A260/280) and similar yield.

### 2.3. Sequencing, Transcriptome Assembly and Analysis

The six samples of total RNA, between 8–12 µg per sample, were provided to a contract sequencing provider, LC Sciences (Houston, TX, USA). The contractor qualified samples using Bioanalyzer 2100 Expert system (Agilent Technologies, Inc., Santa Clara, CA, USA). All samples passed and library preparation was performed according to TruSeq Stranded Total RNA Sample Preparation Guide (Illumina, San Diego, CA, USA, Part # 15031048 Rev. E). Cytoplasmic and mitochondrial ribosomal depletion was performed using Ribo-Zero™ Gold (Human/Mouse/Rat) kit (Illumina, San Diego, CA, USA). Sequencing was performed on the Illumina HiSeq 2000 platform in the 100 bp PE configuration according to the manufacturer’s instructions for running the instrument. Data are available in the NCBI SRA division under BioProject number PRJNA393960. The contractor mapped raw reads to the *S. invicta* genome version Solenopsis_invicta.GCA_000188075.1, available at http://metazoa.ensembl.org/, using TopHat v2.1.0 and then assembled using Cufflinks v2.2.1 [51]. Gene annotations were transferred directly from those provided by the *S. invicta* genome project. We submitted the combined annotated transcriptome for BUSCO [52] analysis to gauge assembly completeness. Cufflinks was used to calculate gene expression levels on a sample-specific basis, expressed using the FPKM measure (Fragments Per Kilobase of transcript per Million mapped reads), which were then used to perform statistical tests of differential expression: transcripts having FDR-adjusted *p*-values (i.e., *q*-values) not exceeding 0.05 were flagged as being differentially expressed. The arithmetic means of bioreplicate-specific FPKM values were calculated for both L4 and WP samples, and binary logarithms of the ratio of these amounts were used to connote the magnitude and direction of change in expression. Visualization of this dataset was performed using R (3.3.0) and RStudio (1.0.136) (R Consortium, Boston, MA, USA), and the R packages data.table (1.10.4)m dplyr (0.7.2), ggplot (2.2.1) [53], and png (0.1–7). The assembled transcriptome was annotated using the Pfam, Gene Ontology (GO) and KEGG databases: the longest amino acid sequence identified among all six reading frames for each transcript was compared with the Pfam-A database using HMMER3 with default parameter settings [54]. GO annotations were extracted from associated Pfam records for each of the ontology’s three aspects: biological process, cellular component, and molecular function. KEGG entries were gleaned via records for *S. invicta* genes previously identified in the organism’s genome annotation project [7,22] that corresponded to differentially expressed transcripts identified in this study. Such entries were organized into pathway-specific modules and manually inspected for trends in expression patterns. Differentially expressed transcripts likely to be associated with ten broadly defined functions/systems having potential for success in downstream, RNAi-based applications were selected and tabulated: digestion, nutrient storage, detoxification, muscle, cuticle, neurons, cell regulation and signaling, hormone, fatty acid metabolism, and immunity. Caste determination genes were also inspected with an expectation of little to no variability.

## 3. Results

### 3.1. Sequencing, Assembly, Differential Expression Analysis and Functional Analysis

A combined total of 59,982,190,474 bases were sequenced from 296,941,537 read pairs using an Illumina Hi-Seq instrument by the commercial sequencing facility (LC Sciences, Houston, TX, USA). Assembly statistics are shown in Table 1. Using the *S. invicta* genome assembly (GCA_000188075.1) as a reference, transcriptome assembly resulted in a total of 156,511,217 bases assembled into 97,078 transcripts. The assembly contained 4150 (94%) complete (of which 2712 appear to be duplicated) and 215 fragmented (4.9%) BUSCO orthologs; we assessed completeness against the insecta_odb9 dataset, for Insecta. Only 50 (1.1%) were not recovered (data not shown) [52]. 

Differential expression analysis identified 2614 DE transcripts (2.7% of total), 1603 of which were ascribed annotations via the pre-existing genome assembly. 362 were annotated as uncharacterized or characterized by more than one gene; 60 transcripts were non-coding RNA. 1260 differentially expressed genes differed by >4-fold and were used for KEGG analysis. The relationships between log2FC, average FPKM and *q*-value among transcripts are visualized in Figure 2. Non-differentially expressed log2FC value distributions were normal and centered around zero, whereas differentially expressed transcripts had a bimodal distribution with a larger proportion having negative log2FC values (i.e., higher expression in larvae relative to pupae; Figure 2).

Pfam and GO analysis of the comprehensive transcriptome suggested nothing especially unusual or distinctive about the *S. invicta* gene set relative to those of other insect taxa (Table 2 and Figure 3). KEGG pathway analysis of >4-fold differentially expressed genes revealed that metabolic pathways were more often upregulated in larvae (524 out of 589, or roughly 89%; see Table 3); an overview of the pathways are listed in Table 3, and the entire list is provided in Appendix A. The number of unique genes represented in Table 3 and Appendix A is 360.

### 3.2. Gene Family Analysis by Functional Category

#### 3.2.1. Digestion, Nutrient Storage and Xenobiotic Detoxification

We identified 58 differentially expressed transcripts related to digestive processes. These were generally expressed in high levels in the larval stage and at lower levels in the pupal stage (Table 4). Forty-three differentially expressed transcripts annotated as various peptidases, proteinases and proteases may be used in the insect gut. The two most abundant transcripts were chymotrypsin-like. One differentially expressed secreted salivary peptide was more abundant in larvae and a differentially expressed transcript annotated as salivary plasminogen activator gamma was more abundant in pupae. Three peritrophin transcripts were differentially expressed, two much more abundant in larvae and the other more abundant in pupae. There were seven differentially expressed transcripts related to lipid or lipase processes, four of which were more abundant in larvae.

We tabulated twelve transcripts related to nutrient storage (Appendix A). Seven differentially expressed transcripts were annotated as trehalose transporters. All, except one, were more abundant in larvae. Two hexamerin transcripts were expressed at extremely high levels in the larval stage. One met the criterion selected for statistical significance for differential expression, and while the other was not differentially expressed, it had a greater than two-fold reduction in transcription level in the pupal stage. Two arylphorin transcripts were expressed at high levels in both life stages but were not differentially expressed.

Sixty-four differentially expressed transcripts related to xenobiotic detoxification processes were almost universally downregulated (61 of 64) from larval to pupal stage (Appendix A). There were 46 differentially expressed cytochrome p450 associated transcripts, two were more abundant in pupae. There were four differentially expressed glutathione s-transferase associated transcripts all more abundant in larvae. There was also a “probable phospholipid hydroperoxide glutathione peroxidase isoform X2” transcript decreased in abundance from larval to pupal stage. There were three carboxylesterase transcripts and four E4/FE4 esterases, all more abundant in larvae. Among three ABC transporter transcripts, two were more abundant in larvae and one in pupae.

#### 3.2.2. Muscle, Cuticle and Neuronal Development

The assembled *S. invicta* transcriptome contained 16 differentially expressed transcripts related to muscle development (Appendix A). Differentially expressed transcripts related to actin, myosin and muscle were primarily (12) downregulated in pupae, four were upregulated. Six myosin-related transcripts were differentially expressed. Four were more abundant in larvae and two, annotated as “unconventional myosin-XV” and “unconventional myosin XVIIIa-like” were more abundant in pupae. Seven actin related transcripts were differentially expressed: Six more abundant in larvae and one more abundant in pupae.

We identified 28 differentially expressed transcripts are related to cuticle development (Appendix A); 12 were more abundant in larvae and 16 in pupae. Four transcripts related to chitinase were differentially expressed, with three being more abundant in larvae and one more abundant in pupae. One transcript related to elastin was differentially expressed, with greater abundance in the larvae. A calphotin transcript was differentially expressed, with greater abundance in the pupae.

Transcripts related to neuron activity or neuronal development were differentially expressed as shown in Appendix A. Of the 17 found, three were synaptic vesicle glycoproteins, all more abundant in larvae. A “neuronal acetylcholine receptor subunit alpha-7-like” and “alpha-2-like”, and a “mesencephalic astrocyte-derived neurotrophic factor homolog” were significantly more abundant in larvae. A neurotactin, a neuromodulin, three semaphorins, a synapsin, two synaptotagmin transcripts were upregulated in pupae.

#### 3.2.3. Cell Regulation, Hormone Signaling and Fatty Acid Metabolism

Twenty-eight transcripts related to cell regulation and signaling were differentially expressed (Appendix A). Fifteen histone transcripts were differentially expressed, with all but one more abundant in larvae than pupae. Two histone-lysine *N*-methyltransferases were more abundant in larvae. Among six differentially expressed elongation factors, five were more abundant in larvae. 

Five transcripts related to hormone signaling were differentially expressed (Appendix A). Differentially expressed transcripts annotated as a juvenile hormone epoxide hydrolase 1-like, growth hormone-inducible transmembrane protein-like, and a broad complex core protein isoform transcript were expressed more abundantly in larvae. One prohormone-2-like and a “probable nuclear hormone receptor HR3” were more abundantly expressed in pupae.

Forty-eight transcripts associated with fatty acid and Coenzyme A (CoA) metabolism were differentially expressed (Appendix A), with the majority (42) being more abundant in larvae. The transcripts more abundant in pupae included “elongation of very long chain fatty acids protein 4-like, partial”, “acyl-CoA synthetase short-chain family member 3, mitochondrial”, “long-chain fatty acid transport protein 4-like”, “fatty acid synthase-like”, and two “fatty acyl-CoA reductase 1-like, partial” that were exclusively found in the pupa samples.

#### 3.2.4. Immunity and Caste Determination

Eight transcripts related to immune processes were differentially expressed (Appendix A). Two toll-like receptor transcripts were differentially expressed, with one being more abundant in the larval stage and the other more abundant in the pupal stage. One “phenoloxidase 2” transcript was also more abundant in the larval stage.

Thirty-one transcripts related to caste determination were observed (data not shown). Of these, 20 transcripts were annotated as yellow proteins; three were differentially expressed, with two more abundant in larvae and one more abundant in pupae. Eleven transcripts were annotated as royal jelly proteins, but none were differentially expressed. Overall, a trend of upregulation of yellow and royal jelly proteins from the larval to pupal stage was observed, although in most cases it was not statistically significant.

## 4. Discussion

The present study utilized high-throughput sequencing to identify hundreds of physiologically relevant transcripts present in two juvenile life stages of the red imported fire ant, *Solenopsis invicta*. We sampled RNA from three individual colonies and sequenced total RNA from each sample (Table 1). While our sample groups were not significantly different (Student *t*-test, paired, two-tailed *p* > 0.1), fewer bases and reads were obtained from the samples from colony 3 (samples C03L4 and Z03wp); we attributed this to random sample processing variation. Large scale differences in developmentally related transcripts were seen between larval and pupal stages. Our expectation was to find abundant transcripts in larva samples associated with digestion of food if the fourth instar larva was indeed the colonies’ community gut. These transcripts should encode enzymes to break down complex proteins, carbohydrates, and fats; nutrient storage and transport genes, and enzymes to detoxify complex ingested molecules. Transcripts encoding genes associated with nutrient metabolism should also be abundant in larvae. Larval transcripts greater than four-fold more abundant were represented in nearly every metabolic pathway (Table 3), while genes upregulated in pupae were proportionally greater in fatty acid metabolism and biosynthesis, purine metabolism, peroxisome and cellular signaling, and only represented in 24 of the 78 dominant pathways (those represented by 4 or more genes in Appendix A). Transcripts associated with cellular processes and muscles are frequently used as reference genes for gene expression studies. Interestingly, some of these “housekeeping” genes were differentially expressed in our analysis, including multiple histones and actin genes (Appendix A). BUSCO analysis demonstrated genes expected to be present in insects in general, and in ants in particular, were captured in these sequencing data, which suggests that the global assembly of these transcriptomes comprises a subset of genes present in this species.

Many physiological processes change between the larval and pupal life stages. The identification of transcripts related to these processes contributes to an increased understanding of *S. invicta* developmental biology, enabling future RNAi gene targeting. The data presented here support many observations regarding the biological roles of developmental stages in the ant colony, or superorganism; a comparative abundance of digestive, nutrient storage and detoxification genes were differentially and abundantly transcribed in fourth instar larvae. In addition, genes associated with all metabolic pathways were differentially transcribed; 308 of the 360 unique genes listed in Appendix A were more abundant in larva samples (85.6%) This corroborates observations and research conclusions that the larval stage provides communal digestive functions for the colony [1,27]. *S. invicta* larvae digest food, often rich in protein, and regurgitate some of it to be shared with other members of the colony. In addition, larvae also need to develop and store energy and nutrients for use during pupation. Digestion occurring in the larval stage requires an abundance of digestive enzymes; indeed, high expression of digestive enzyme transcripts was observed in the larval stage, including chymotrypsin, alpha-amylase, other various proteinases, proteases, peptidases, and salivary peptides (Table 4 and Appendix A). These enzymes, as well as peritrophin, a structural component of the gut’s peritrophic matrix [55], were present in the pupal stage at much lower levels, likely due to decreased digestive activity in the pupal stage. Interestingly, our results both support and contrast RNAi research linking chemosensory proteins and juvenile developmental processes; some of the transcripts encoding chemosensory proteins, fatty acid synthase, and housekeeping standard genes were identified in our data [44] (also see [21]). These complex relationships are outside of the scope of our present study, and deserve further analysis to decipher.

One of the strategies for storing energy and nutrients during the larval stage is expression of storage proteins [56]. These storage proteins can later be degraded in the pupal stage to supply amino acids for developmental processes. Hexamerins function as insect storage proteins with importance in colony founding, egg development and brood nourishment, as has been documented in *Camponotus festinatus* [57,58]. More recent investigations suggest that hexamerins may also be involved in insect immunity, hormone transport and cuticle formation [59]. Hexamerin associated transcripts were highly expressed in the larval stage. They were less abundant but still present in large amounts in the pupal stage (Appendix A). Arylphorins constitute another category of storage proteins, similar in structure to hexamerins but containing more aromatic amino acid residues [60]. Two aryrlphorin transcripts were expressed at high levels in both life stages, but expression levels did not statistically differ. It is logical for high levels of transcription to occur during the larval stage, but it appears the pupal stage is also actively transcribing hexamerin and arylphorin storage proteins. Not only were digestive and storage proteins (Table 4 and Appendix A, respectively) more abundant in larvae, but they were among the most abundantly transcribed genes among all detected. Interestingly, fatty acid processing appears to be important in both the larval and pupal stage. Both fatty acid degradation and biosynthesis transcripts are abundant in the fourth instar larva, while several fatty acid synthase transcripts are highly abundant in pupae (Appendix A). This could be in preparation for synthesis of long chain hydrocarbons used for the complex communications between adults [21,61].

Trehalose is an important insect blood sugar regulated by trehalase. Trehalase and various trehalose transporters were differentially expressed, being transcribed at higher levels in the larval stage. Trehalase has been associated with numerous insect processes [62], such as sugar metabolism and growth, metamorphosis and reproduction, flight, chitin synthesis and stress recovery. Thus, trehalase may be considered a nutrient storage and metabolism enzyme; we have placed it in the list of digestive enzymes because the diet of *S. invicta* is primarily insects. The downregulation of trehalase, numberous trehalose transporters and an alpha,alpha-trehalose-phosphate synthase, indicate a reduction in this pathway in the pupal stage, probably in response to the cessation of feeding processes during pupation.

The assembled *S. invicta* transcriptome contains many differentially expressed cytochrome p450s, carboxylesterases, glutathione s-transferases, ABC transporter proteins and peroxiredoxins. In almost all cases, the larval stage had a higher expression level than the pupal stage (Appendix A), suggesting a decrease in xenobiotic detoxification machinery after the final larval metamorphosis. *S. invicta* is omnivorous and much of the colony’s food consists of other insects, either scavenged or prey [1]. Many insects produce or sequester toxic compounds to deter predation. Larvae are therefore exposed to foodborne toxins, including entomopathogens and pesticides, in addition to naturally occurring metabolic byproducts. Abundant expression in the larval stage to combat xenobiotic compounds from food and a subsequent reduction in the non-feeding pupal stage is logical. Several of these enzyme families have been associated with increased tolerance of insecticides and/or insecticide resistance. Understanding the repertoire of xenobiotic detoxification machinery and their expression levels can give us insight into *S. invicta*’s susceptibility to insecticidal compounds, as well as ability to develop resistance. Here, high levels of larval expression of several cytochrome p450, E4 esterase and glutathione S-transferase transcripts were observed. While E4 and FE4 esterases are generally categorized as detoxifying enzymes (see [63]), recent research indicates that esterase FE4 may have a role in pheromone biosynthesis in hymenoptera [64]. The sizeable larval xenobiotic repertoire, compounded with the difficulty of reaching the larvae within the subterranean nest, may reduce efficacy of insecticide application. Foragers are colony members that are most likely to come in contact with insecticides, however they are typically the oldest and most expendable members of a colony. 

Two muscle related transcripts not annotated as actin- or myosin-related were differentially expressed; one was annotated as a “muscle-specific protein 20” and the other as “muscle LIM protein Mlp84B-like isoform X1”, and both were more abundant in larvae. The first was identified in KEGG (http://www.genome.jp/dbget-bin/www_bget?soc:105208199, accessed 4 May 2018) as transgelin, a protein characterized in human fibroblasts [65] but also found in the trophallactic fluid of ants [17]. Mlp84B is important in maintaining muscle integrity in *Drosophila melanogaster* [66]. The specific function of these muscle related genes in ants is unknown but worthy of examination, and these could be candidates for ant RNAi studies.

As expected, cuticle related proteins were differentially expressed across the two life stages, highlighting the complexity of insect cuticle development [67]. More intensive study of ant cuticular proteins is needed, and because the insect cuticle is so vital to survival these transcripts may be candidate RNAi targets. Chitinase enzymes degrade chitin during the molting process, which occurs during normal larval to pupal development [68]. Differential expression of some molting related genes was expected, but relatively few characterized transcripts were captured by our analyses (Appendix A); our specimens were not apparently molting (visual inspection). Chitinase is also important in digestive processes to degrade chitin in foodstuffs, such as insect prey [69]. Chitinase being differentially expressed, with greater abundance in the larval stage relative to pupal, could result from differences in digestive processes and/or be related to structural modification of chitin within the organism. 

Upregulation of calphotin suggests that eye development takes place, at least in part, during the pupal stage [70,71]. Calphotin is a gene expressed in the soma and axons of insect photoreceptor cells early in their development (Ballinger 1993). The calphotin gene encodes a calcium-binding protein. While not present in the rhabdomeres [71] the calphotin protein is important to rhabdomere development [72]. Because the rhabdomeres of *S. invicta* are not developed until the pupa-adult stage, increased expression (greater than six-fold) of this gene in the pupa samples supports the assumption that this gene functions in a manner in ants similar to the *Drosophila* model organism.

Many neuronal components were differentially expressed between *S. invicta* larvae and pupae (Appendix A). The larval stage contained three differentially expressed synaptic vesicle glycoproteins, as well as neuronal acetylcholine receptor subunit alpha-2-like and mesencephalic astrocyte-derived neurotrophic factor transcripts. The pupal stage had 12 upregulated transcripts related to nerve synapses and synaptic transmission, such as semaphorins, synapsins, neuromodulin and synaptogyrin. One neuropeptide in particular, “neuropeptide-3” was expressed in extremely high amounts in the pupal stage. This transcript encodes a short 80 amino acid neuropeptide. Many roles have been associated with neuropeptides in insects (reviewed in [73]), and future investigation of *S. invicta*’s differentially expressed neuropeptides is warranted, both for scientific understanding and RNAi targeting. The neural genes upregulated in the pupa stage probably indicate development of adult communication mechanisms.

We were surprised by the 28 differentially expressed histone related transcripts in our data set, and this motivated us to include the category cell regulation and signaling in our analysis (Appendix A). We can only hypothesize that the overall metabolic activity apparently occurring in the late larval stage of *S. invicta* necessitates the production of new or replaced histones. We also noted that several differentially expressed transcripts annotated as hormones appeared in our data (Appendix A). While we expected juvenile hormone associated transcripts in the larval samples based on basic juvenile insect physiology, the other “hormones” are not well characterized in ants or other insects, and may warrant future investigation. Because our study organisms were not intentionally immune-challenged, the small number of immunity related transcripts that we identified (Appendix A) may represent a baseline presence in those samples. Two toll-like receptor transcripts were noted, because the toll pathway is important in the innate immune system. Two transcripts, one annotated as “endoribonuclease Dcr-1-like, partial” and the other as “protein argonaute-2”, were more abundant in the larval stage. These two transcripts may play an important role in the RNAi pathway.

Yellow and royal jelly proteins may play a role in hymenoptera caste determination. Drapeu et al. 2006 [74] characterized developmental, sex, and caste specific expression patterns of yellow and royal jelly genes in *Apis mellifera*. In addition, *A. mellifera* queen larvae are fed large amounts of royal jelly, which is produced by the workers [75]. The *S. invicta* transcriptome contains many yellow and royal jelly proteins; however, most were not differentially expressed in our experiment (data not shown). Because this study was based only on samples of one caste, workers, we would not expect dramatic transcriptional variance between stages. Nonetheless, the roles of these genes are not well characterized in ants; exploring the role of yellow and royal jelly genes in ant polyphenism remains an interesting area for future research.

## 5. Conclusions

Targeting *S. invicta* larvae with biopesticides, such as RNAi, may prove to be a more efficacious mechanism than broad-spectrum synthetic chemical pesticides for several reasons. First, as we have learned with previous control attempts using insecticides across large areas, specificity matters. Off-target effects occurred, killing many unintended organisms [5]. RNAi can be used to knock down a particular gene’s transcription level. Often, RNAi gene targets do not broadly occur in organisms that may encounter them, and if they do, they should vary enough in sequence similarity to prevent successful gene knockdown. Second, it enables circumvention of standing xenobiotic defense capabilities in larvae. Using the RNAi pathway to knock down a gene’s expression level has been used in multiple organisms, including *S. invicta* individuals [42,43,44,45], and should also work in the superorganismal context. Third, it allows specifically targeting larvae, which are vital to the colony’s digestion capabilities. By using an RNAi bait, we can deliver RNAi technology to larvae via foragers. Foragers may not be killed and should continue to deliver additional RNAi-laced bait. Targeting the larvae with RNAi could be an effective control strategy because of the larvae’s important role as a “communal gut”. The effects could have numerous impacts across the colony. Direct effects of RNAi may include: an increase in larval mortality, a lower percentage of larvae proceeding to pupation, an increase in time preparing for pupation, an increase in pupa mortality, and lower adult fitness. Our initial evaluation of the transcripts identified in this analysis point to some interesting target candidates: the transcript associated with larva salivary activity, LOC105196698 and the larval peritrophin transcripts, LOC105194610 and LOC105196420 appear promising. RNAi targeting the larval stage may indirectly disrupt protein availability for all colony members, most importantly the queen, resulting in a reduction in egg laying. Additionally, oviposition is primarily regulated by fourth instar larvae [76], so we speculate that this stage constitutes a critical “weak spot” in colony vitality. We plan to develop colony-level bioassays for RNAi experiments in *S. invicta*, requiring targets, positive controls, and negative controls. This is not a trivial challenge.

Our study identified differentially expressed genes in the pupal stage that could also be critical to colony vitality. If RNAi bait is delivered to the larvae, will it persist long enough to knock down incipient pupal developmental genes? If so, then targeting the pupae could also be a viable RNAi application strategy. Vital developmental processes occur in the pupal stage, along with the final molting process resulting in eclosion of a mature adult. Thus, the colony could be disabled by either interference with processes specific to the late stage larva or the pupa. The transcriptome data generated in this study constitute an important resource in identifying safe and effective RNAi targets for *S. invicta*. Such key factors to be considered in this decision making include whether to select targets on the basis of their role in developmental processes, protein product half-life, transcription levels, as well as gene copy number. This analysis reveals many potential targets; identifying those that are vital to colony survival awaits further study.

## Figures and Tables

**Figure 1 insects-09-00185-f001:**
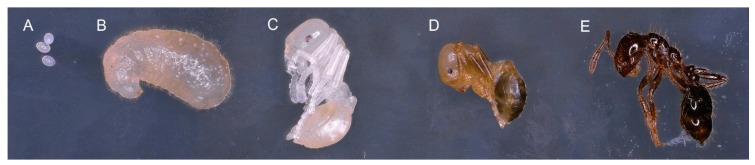
Developmental stages of the *S. invicta* worker caste: (**A**) egg; (**B**) larva; (**C**) early stage (white) pupa; (**D**) late stage pupa or pharate adult; (**E**) adult. RNA samples were extracted from representatives similar to (**B**,**C**).

**Figure 2 insects-09-00185-f002:**
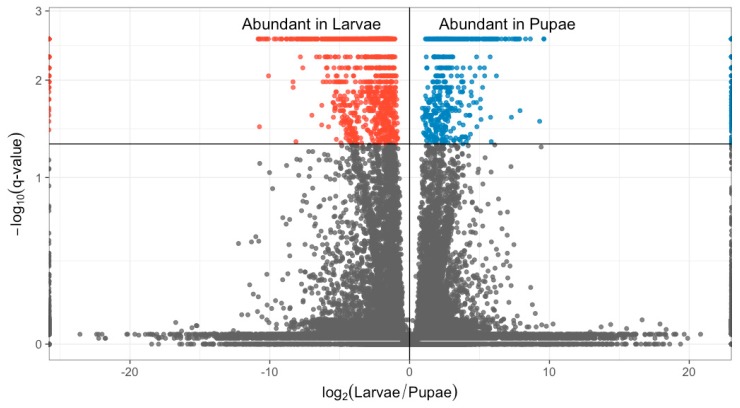
Visualization of transcript expression: data relating to differentially expressed transcripts are shown in red and blue, non-differentially expressed transcripts in black. Volcano plot displays the relationship between fold change and *q*-values.

**Figure 3 insects-09-00185-f003:**
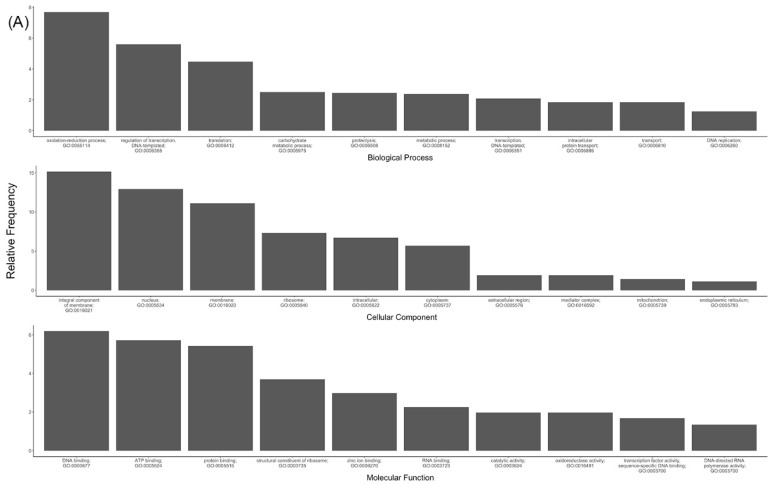
Ten most common Gene Ontology (GO) molecular function, cellular component and biological process categories of *S. invicta* transcripts: (**A**) full ontology, (**B**) slim ontology.

**Table 1 insects-09-00185-t001:** Individual sample sequencing statistics. Samples with names ending in wp are pupal samples, those with names ending in L4 are larvae. Numerals in sample names designate source laboratory colony.

Sample	Total Read Bases (bp)	Total Reads	GC (%)	AT (%)	Q20 (%)	Q30 (%)
X01wp	11,021,733,676	109,126,076	49.37	50.63	93.77	89.38
Y02wp	11,915,906,472	117,979,272	49.03	50.97	92.97	88.54
Z03wp	7,833,054,394	77,554,994	48.38	51.62	94.65	90.29
A01L4	10,596,167,954	104,912,554	48.95	51.05	93.83	89.57
B02L4	10,050,206,394	99,506,994	46.85	53.15	93.7	89.45
C03L4	8,565,121,584	84,803,184	46.39	53.61	94.88	90.78

**Table 2 insects-09-00185-t002:** Top 20 most numerous PFamilies.

Quantity	Symbol	PFamily Annotation	PFamily ID
1243	RVT_1	Reverse transcriptase (RNA-dependent DNA polymerase)	PF00078.25
849	Pkinase	Protein kinase domain	PF00069.23
823	Pkinase_Tyr	Protein tyrosine kinase	PF07714.15
676	rve	Integrase core domain	PF00665.24
641	zf-C2H2	Zinc finger, C2H2 type	PF00096.24
625	zf-C2H2_4	C2H2-type zinc finger	PF13894.4
512	RRM_1	RNA recognition motif. (a.k.a. RRM, RBD, or RNP domain)	PF00076.20
490	Ig_3	Immunoglobulin domain	PF13927.4
478	WD40	WD domain, G-beta repeat	PF00400.30
473	zf-H2C2_2	Zinc-finger double domain	PF13465.4
472	I-set	Immunoglobulin I-set domain	PF07679.14
462	MFS_1	Major Facilitator Superfamily	PF07690.14
455	Ig_2	Immunoglobulin domain	PF13895.4
449	ig	Immunoglobulin domain	PF00047.23
426	7tm_6	7tm Odorant receptor	PF02949.18
420	p450	Cytochrome P450	PF00067.20
400	V-set	Immunoglobulin V-set domain	PF07686.15
345	Transposase_1	Transposase (partial DDE domain)	PF01359.16
334	ANAPC4_WD40	Anaphase-promoting complex subunit 4 WD40 domain	PF12894.5
327	DDE_3	DDE superfamily endonuclease	PF13358.4

**Table 3 insects-09-00185-t003:** Overview of pathways represented by >4-fold differentially expressed genes (DEG) found in larva and pupa *Solenopsis invicta* samples. Details provided in Appendix A.

Category	Subcategory	Number	>DEG Larvae	>DEG Pupae
Metabolic pathways				
	Amino acids	67	67	0
	Fats/Lipids	63	47	16
	Nucleic acids	39	33	6
	Carbon metabolism	36	36	0
	Carbohydrates	35	35	0
	Detoxification	33	31	2
	Vitamins	19	19	0
	Amino sugar and nucleotide sugar metabolism	14	14	0
	Pyruvate metabolism	11	11	0
Cellular processes				
	Signaling pathways	42	23	19
	Membrane trafficking	16	14	2
	Genetic information processing	9	5	4
	Apoptosis	6	5	1
Organelle Biosystems				
	Peroxisome	29	25	4
	Phagosome	17	16	1
	Protein processing in endoplasmic reticulum	15	15	0
	Lysosome	14	14	0
	Ribosome related	4	4	0
Biosynthesis pathways				
	Biosynthesis of amino acids	23	23	0
	Fatty acid biosynthesis	18	10	8
	Insect hormone biosynthesis	13	12	1
	Terpenoid backbone biosynthesis	10	10	0
Degradation pathways				
	Amino acid degradation	28	28	0
	Fatty acid degradation	18	18	0
	Glycosaminoglycan degradation & other glycan degradation	7	6	1
	RNA degradation	3	3	0
	Totals	589	524 (89%)	65 (11%)

**Table 4 insects-09-00185-t004:** Differentially expressed transcripts likely to be associated with digestion. Larvae and pupae values are measured in averaged FPKM. *q* is corrected for multiple comparisons. Negative numbers, shaded in blue fading to yellow, represent upregulation in larvae; positive numbers, shaded orange to red, represent upregulation in pupae.

Gene ID	Annotation (Predicted)	Larvae	Pupae	Binary Log (Fold Change)	*p*	*q*
LOC105199117	chymotrypsin-1-like	14,120.80	7.70	−10.84	5.00 × 10^−5^	2.75 × 10^−3^
LOC105194373	phospholipase A1-like, partial	383.05	0.22	−10.77	1.05 × 10^−3^	3.38 × 10^−2^
LOC105199115	chymotrypsin-1-like	8648.28	5.45	−10.63	5.00 × 10^−5^	2.75 × 10^−3^
LOC105193961	chymotrypsin-1-like	26,016.40	18.53	−10.46	5.00 × 10^−5^	2.75 × 10^−3^
LOC105198380	lipase 3-like, partial	1058.65	0.97	−10.09	2.00 × 10^−4^	8.84 × 10^−3^
LOC105193957	chymotrypsin-1-like	510.44	0.81	−9.30	5.00 × 10^−5^	2.75 × 10^−3^
LOC105208099	lipase 3-like	368.52	0.65	−9.15	5.00 × 10^−5^	2.75 × 10^−3^
LOC105198043	alpha-amylase	2527.41	5.70	−8.79	5.00 × 10^−5^	2.75 × 10^−3^
LOC105194610	peritrophin-1-like	4362.25	11.73	−8.54	5.00 × 10^−5^	2.75 × 10^−3^
LOC105193995	chymotrypsin-2-like	528.99	1.64	−8.33	3.00 × 10^−4^	1.23 × 10^−2^
LOC105196698	probable salivary secreted peptide	463.02	1.66	−8.12	1.60 × 10^−3^	4.76 × 10^−2^
LOC105199102	venom metalloproteinase 3-like	240.80	1.27	−7.57	5.00 × 10^−5^	2.75 × 10^−3^
LOC105196175	zinc carboxypeptidase-like	2455.41	18.21	−7.08	5.00 × 10^−5^	2.75 × 10^−3^
LOC105193306	chymotrypsin-1-like	186.87	1.76	−6.73	5.00 × 10^−5^	2.75 × 10^−3^
LOC105196420	peritrophin-1-like	1162.08	15.00	−6.28	5.00 × 10^−5^	2.75 × 10^−3^
LOC105200093	chymotrypsin-2-like, partial	639.60	10.38	−5.95	1.50 × 10^−4^	6.96 × 10^−3^
LOC105197108	aminopeptidase N, partial	15.65	0.26	−5.91	1.50 × 10^−4^	6.96 × 10^−3^
LOC105205908	chymotrypsin-2-like, partial	216.51	4.93	−5.46	5.00 × 10^−5^	2.75 × 10^−3^
LOC105207761	anionic trypsin-2-like	100.83	3.36	−4.91	5.00 × 10^−5^	2.75 × 10^−3^
LOC105200277	inducible metalloproteinase inhibitor protein-like	27.20	1.03	−4.72	6.50 × 10^−4^	2.28 × 10^−2^
LOC105200273	chymotrypsin inhibitor-like	1284.77	56.33	−4.51	5.00 × 10^−5^	2.75 × 10^−3^
LOC105200221	retinoid-inducible serine carboxypeptidase-like	169.37	7.85	−4.43	5.00 × 10^−5^	2.75 × 10^−3^
LOC105200003	glutamyl aminopeptidase isoform X3	123.56	6.06	−4.35	5.00 × 10^−5^	2.75 × 10^−3^
LOC105193273	chymotrypsin-2-like	156.98	8.14	−4.27	5.00 × 10^−5^	2.75 × 10^−3^
LOC105195679	serine proteinase stubble	35.89	2.16	−4.05	5.00 × 10^−5^	2.75 × 10^−3^
LOC105203057	endoplasmic reticulum metallopeptidase 1-like isoform X4	21.71	1.57	−3.79	1.50 × 10^−4^	6.96 × 10^−3^
LOC105205350	xaa-Pro dipeptidase	103.15	7.71	−3.74	5.00 × 10^−5^	2.75 × 10^−3^
LOC105200222	retinoid-inducible serine carboxypeptidase-like	97.19	7.46	−3.70	5.00 × 10^−5^	2.75 × 10^−3^
LOC105203057	endoplasmic reticulum metallopeptidase 1-like isoform X2	81.14	7.32	−3.47	5.00 × 10^−5^	2.75 × 10^−3^
LOC105206478	aminopeptidase N-like, partial	107.22	10.36	−3.37	5.00 × 10^−5^	2.75 × 10^−3^
LOC105203676	dipeptidyl peptidase 3 isoform X2	189.57	19.73	−3.26	5.00 × 10^−5^	2.75 × 10^−3^
LOC105200810	digestive cysteine proteinase 1	1,995.67	217.85	−3.20	5.00 × 10^−5^	2.75 × 10^−3^
LOC105206533	aminopeptidase N-like, partial	122.02	13.78	−3.15	1.00 × 10^−4^	4.96 × 10^−3^
LOC105203240	trehalase-like	25.37	3.41	−2.90	5.00 × 10^−5^	2.75 × 10^−3^
LOC105196148	cytosolic non-specific dipeptidase	217.85	34.02	−2.68	5.00 × 10^−5^	2.75 × 10^−3^
LOC105195467	xaa-Pro aminopeptidase 1	95.29	15.49	−2.62	5.00 × 10^−5^	2.75 × 10^−3^
LOC105196184	signal peptidase complex subunit 1	98.78	20.91	−2.24	5.00 × 10^−5^	2.75 × 10^−3^
LOC105204623	mitochondrial-processing peptidase subunit beta	163.92	35.95	−2.19	5.00 × 10^−5^	2.75 × 10^−3^
LOC105197517	aminopeptidase N	11.48	2.80	−2.04	5.00 × 10^−5^	2.75 × 10^−3^
LOC105198213	signal peptidase complex subunit 3	300.89	74.52	−2.01	5.00 × 10^−5^	2.75 × 10^−3^
LOC105199614	puromycin-sensitive aminopeptidase isoform X1	61.68	16.73	−1.88	5.00 × 10^−5^	2.75 × 10^−3^
LOC105200356	signal peptidase complex catalytic subunit SEC11A	73.13	20.48	−1.84	5.00 × 10^−4^	1.85 × 10^−2^
LOC105198128	probable signal peptidase complex subunit 2	100.86	33.86	−1.57	1.00 × 10^−4^	4.96 × 10^−3^
LOC105193333	prolyl endopeptidase-like, partial	87.30	32.62	−1.42	6.00 × 10^−4^	2.14 × 10^−2^
LOC105201160	mitochondrial-processing peptidase subunit alpha	51.37	20.68	−1.31	5.00 × 10^−5^	2.75 × 10^−3^
LOC105201089	putative phospholipase B-like lamina ancestor	214.83	101.38	−1.08	5.00 × 10^−5^	2.75 × 10^−3^
LOC105200792	disintegrin and metalloproteinase with thrombospondin motifs 7-like	3.97	11.35	1.52	5.00 × 10^−5^	2.75 × 10^−3^
LOC105197096	peritrophin-1-like, partial	13.82	45.98	1.73	1.00 × 10^−4^	4.96 × 10^−3^
LOC105193940	disintegrin and metalloproteinase domain-containing protein 11, partial	2.61	10.41	2.00	6.00 × 10^−4^	2.14 × 10^−2^
LOC105193191	sn1-specific diacylglycerol lipase beta-like	2.34	10.15	2.12	4.50 × 10^−4^	1.70 × 10^−2^
LOC105205498	dipeptidase 1-like	1.02	5.77	2.50	1.05 × 10^−3^	3.38 × 10^−2^
LOC105202515	salivary plasminogen activator gamma	4.49	30.83	2.78	5.00 × 10^−5^	2.75 × 10^−3^
LOC105202515	salivary plasminogen activator gamma	1.69	22.46	3.73	5.00 × 10^−5^	2.75 × 10^−3^
LOC105197409	membrane metallo-endopeptidase-like 1 isoform X4	5.05	71.25	3.82	5.00 × 10^−5^	2.75 × 10^−3^
LOC105198885	serine proteinase stubble	0.71	22.38	4.98	5.00 × 10^−5^	2.75 × 10^−3^
LOC105194705	carboxypeptidase B-like	4.20	145.31	5.11	8.50 × 10^−4^	2.84 × 10^−2^
LOC105198972	phospholipase B1, membrane-associated-like, partial	-	1.83	inf	5.00 × 10^−5^	2.75 × 10^−3^
LOC105205146	phospholipase B1, membrane-associated-like, partial	-	2.47	inf	5.00 × 10^−5^	2.75 × 10^−3^

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
