# Peer review of "Differential Gene Expression in Red Imported Fire Ant (Solenopsis invicta) (Hymenoptera: Formicidae) Larval and Pupal Stages"

_insects, 2018, doi:10.3390/insects9040185_

Reviewer 1 Report

In the present manuscript, the authors investigate differential gene expression between larvae and pupae of the red imported fire ant. Solenopsis invicta is a very successful invasive species that is doing strong economic and ecological damages to the US. There has already been numerous attempt to control the expansion of S. invicta populations and indeed more information are needed in order to introduce more sustainable and ecologically friendly control methods.

The authors use RNA sequencing to find candidate genes (separated in multiple functional groups) that could be of potential use for future ecologically friendly methods.

The present work is of interest for the ever-growing community interested in pest control, and especially Solenopsis invicta. However, I believe that some changes could benefit the manuscript prior to publication, especially when it comes to the hypothesis behind the work. The introduction is well written but could perhaps use more information regarding ecological pest controls and RNAi. Also, there are not many missing references regarding previous work looking at S. invicta differential expression (I am not sure I know of any RNA sequencing datasets out there but there are surely plenty of microarrays studies). How does your study differ from what has been done previously?

Please state your hypothesis clearer as well, and expectations early on in the manuscript. How can your results be used widely? How do your results fit what we previously know of S. invicta? How can they be useful for the eco-friendly management methods? This last question especially needs to be answered more clearly in the manuscript.

I am puzzled regarding the use of a genome-guided transcriptome rather than realigning to the genome directly and use the expression count from genes annotated on the genome. It results in 90000 transcripts which is way too many for a transcriptome (considering S. invicta must have around 15000 genes). The results are not particularly well explained and neither the methods, which differential expression software did you use? why is it based on FPKM? I am rather perplexed but I would like to give a chance to the authors to explain the methods more accurately.

Also, there are an incredibly high number of tables (very long tables) in this paper, I appreciate that the authors do provide all results but I was wondering if it was possible to leave the tables in the supplementary materials instead (it was rather long when I printed it) – depending on the journal policy of course.

Introduction

L 38. Could perhaps mention if reasons for the unsuccessful attempts were known, how can we improve the methods based on earlier attempts?

L 39.  What types of ecological methods can be used as pest controls? What kind of data is needed in order to establish successful protocols?

L 44. Perhaps an introduction to what is RNAi, how does it function and if it has been successful for other pest management. Especially given the paragraph starting L 67.

L 72. I am not sure I entirely agree, if the product was lethal immediately – yes it would kill only foragers, however, foragers could also bring the product to the colony to feed it to other ants/stages.  And how would you plan to target pupae and larvae which are kept in isolation inside the colony? Could you please provide more elements to the management plan?

L 75. Could you please explain in more details why these two stages are important and how they fit your hypothesis. Right now it is not clear on how one can use these lists of DEGs and create a eco-friendly pest control plan and how you plan to use these gene lists to conduct RNAi.

Please state your hypothesis more clearly

Materials and Methods

L 95. It is unclear why your immobilize the adults and replace them, please explain

L100. How many individuals were pooled per sample? did they all come from the same colony? It is really unclear

L 120. I am not understanding why a transcriptome was created when you have a fully annotated and available genome, why not running a software such as RSEM that will give gene expression level by taking directly the gene annotation from the genome. Then the BUSCO results become irrelevant because you already have gene annotations.

L122. Which software did you use for differential expression? EdgeR or DESeq (the two most commonly found) only use raw reads and never FPKM.

L 129. Why annotate again? I thought you transferred the gene annotation  from the genome, it is confusing

L 135. Identified before? please cite or refer to

L 137. Why these lists? You never really stated your hypothesis or why you are particularly interested in these categories

Results

L 145. Trivial point but which Illumina instrument? Hi- Next- Seq? These kinds of details are missing

L 148. This is way too many transcripts, you potentially have fragmented transcripts, contaminants (virus. fungi, bacteria). If you are not using the genome as a reference and really want to use a genome based transcriptome you will need to clean your reference transcriptome (open reading frame, contamination ..)

Table 1. I have no idea what this pvalue refer to and I do not understand it, please explain

L 173. what does unusual refer to? what were your expectations?

Figure 2 is not essential in my opinion

L 175. Any stats regarding this result?

L 294. Why looking at caste determination genes? If your main purposes is to find candidate genes that can be used to create an eco-friendly control methods, caste determination genes are out of the scope of this manuscript. Also as you only have worker samples you have no evidence that these particular genes would be caste-biased, there are numerous studies showing that caste-biased genes are always conserved across species.

Perhaps you could add some heatmaps of candidate/interest genes. Right now your list of genes does not allow for a nice visualization of your results.

Discussion

L 303. Billions of bases is not relevant as a quality stat, you could add the coverage or the average number of reads or just simply remove this part of the sentence.

L 305. Is that relevant to the results? Was it controlled during the analysis?

L 307. First time I read anything about expectations (I could have missed it earlier), these expectations need to be clearly cited with the hypothesis.

L 318. Housekeeping genes differentially expressed is not surprising given the fact that larvae and pupae are two entirely different life stages with physiological, and morphological changes (there are plenty of housekeeping genes DE between castes for instance).

L 320. “the majority of genes”  - busco only look at around 5000 genes and only genes conserved across Hymenoptera species (keeping in mind that an ant genome has x3 times more genes). You are not looking at Solenopsis specific genes, for this you would need to realign your transcript to the genome and see how many of them are covered. I disagree with this statement

L 324. This is also way too strong, you cannot make statements about developmental stages when you are looking at one category of pupae and one larvae.

L 326. “robustly transcribed” Where does that come from? what do you refer to?

L 327. You need stats to affirm something like this (I already mention it in the results I think)

A lot of genes are mentioned in the discussion but without any real background information earlier in the manuscript or precise hypothesis about them. The discussion is rather unclear as a result.

Conclusions

L 442. Is RNAi a biopesticides? How can it be used as one?

You are giving a rather long list of candidate genes, Perhaps you would need to cite the most important ones to target that can be used for RNAi pest control methods. Many of the genes you are referring to in the tables are not Solenopsis specific so how can they be used to target specifically the red fire ant? Which ones would actually be lethal to an ant? Can you compare your list with previous successful RNAi control management in other species?

Your RNAi paragraph should be moved to the introduction rather than the conclusion, here I would like to see conclusions about your work and results rather than an explanation on how successful RNAi control can be

L 459. Hypothesis that need to be moved before

Author Response

Please see word document attached.

Reviewer 2 Report

In this manuscript, Allen et al. present the results of a differential transcriptome analysis of the fire ant Solenopsis invicta, where they compare fourth instar larvae and early pupa stages. I think the paper is well written, shows some interesting data that could be of interest to readers and future research and in general the experiments are conducted correctly. Although I cannot really evaluate the bioinformatics, because I feel I don't have the expertise and experience to do so. But at first sight, they seem common practices for this kind of work.

While I think the results do warrant publication, I do have a few remarks and I think there are a few aspects of this research that could have been improved from the start.

First of all, throughout reading the paper, I was wondering why the authors didn't include adults as a developmental stage for transcriptome sequencing. Especially since they put some emphasis in the introduction on the fact that larvae act as a protein digestive organ for the colony and that adults are limited in their digestive capabilities. I think it is a missed opportunity to not include adults here as well and compare genes inolved in these processes. When comparing larvae with pupa, all differences in digestive genes (and many other processes) are most likely just related to the fact that the pupal stage is not feeding (or is in a quasi-dormant state) and could be completely different again in the adult.

Another thing I was wondering was whether the authors manually annotated any of the genes they saw differentially expressed and discussed in this paper, or whether all functions and identification came from mapping these transcripts to the genome. Which might or might not be well-annotated. These days, automatic annotations are definately improving, but from personal experience I can say that manual annotation can still vastly improve this.

One thing I found surprising in the results here was the limited number of hormonal regulation/developmental/molting genes found differentially expressed between both stages. I would expect to see many more here. Do the authors know how this compares with other insect species? Is there similar work done in other species which really compares expression of these genes between larvae and pupae? Of course, the difficulty with working with these genes is their very 'narrow expression window'. These hormones (and genes) are typically upregulated only very shortly somewhere within an instar and rapidly go down again. So it is possible that this signal cannot be detected because of this reason. But if you look at ecdysone for example, the ecdysone hormone is thought to be typically present in pupae throughout a large part of the pupal stage, but much more briefly in larval stages. Nonetheless, there is no sign of any of the ecdysteroid synthesis genes.

The synchronization of L4 larvae was based on morphological characteristics. But is it possible to give some sort of a time window based on those characteristics? How long does the 4th stage last and for how much of that do these larvae have these darkened mandibles and this size for example? Was it a very narrow time window or more broad? Was it late L4, mid L4 or early L4 or is it difficult to distinguish this?

In the discussion, the authors also make the link to potential RNAi applications for the control of S. invicta. While the idea to selectively target larvae or pupae sounds interesting and could definately work, I was wondering whether just targeting the foragers for mortality would really not have a similarly effective outcome and have a very disruptive effect on the whole colony. Also, killing the foragers using an RNAi application would take time anyway. Usually, lethal effects only become visible after several days. Which would give the foragers time enough to also bring the RNAi bait to the colony and the larvae.

Some smaller things:

In table 9, the last gene has no value in the 'pupae' column

Line 403: suggests

Line 409 Drosophila should be italic

Author Response

Please see attached Word document.

Round  2

Reviewer 1 Report

Thank you to the authors for addressing the issues that I raised in my previous review, in relation to these previous comments I am just answering some that still need corrections. Overall, I am happy with the revisions and I would recommend to the editor that the manuscript be accepted after minor corrections (listed below).

1.     I agree with the reviewers that tables after Table 5 should be placed in the supplementary materials

2.     I am concerned that the pick and replace (CO2 based) adult treatments may have affected also the expression of the larvae and pupae. CO2 gazing is widely known to affect gene expression. Could you please indicate if this is a possibility? I have problem visualizing how the treatment has been done, have the larvae and pupae been gazed as well?

3.     Previously point 19. I would like to know if out of all these transcripts you could have potentially miss, and omit to remove from your transcriptome, contamination such as viruses, fungi and bacteria transcripts that are highly common in ant transcriptomes

4.     Previously point 20. Yes please, as I do not really see the point of the p-value and you don’t seem too attach to it, I would suggest to remove it

5.     Figure 2 is fine now, I can suggest to keep the other parts (A, C and D) as supplementary materials if the authors want

6.     Previously point 36. I agree that comparing to one gene is really small (BTW is the previously commercialized RNAi gene found in your dataset). Datamining is one thing and I agree that you are providing a really good dataset for future studies (you or someone else for that matter), but I would have liked to see some interesting genes to be pointed out in the discussion. Which genes do YOU think are the best candidates for future studies? 

Author Response

Please see attached Word document. Thank you for your assistance!
